# Pit One Against Many: Leveraging Attention-head Embeddings for Parameter-efficient Multi-head Attention

**Huiyin Xue** and **Nikolaos Aletras**
Department of Computer Science, University of Sheffield
United Kingdom
{hxue12, n.aletras}@sheffield.ac.uk

## Abstract

Scaling pre-trained language models has resulted in large performance gains in various natural language processing tasks but comes with a large cost in memory requirements. Inspired by the position embeddings in transformers, we aim to simplify and reduce the memory footprint of the multi-head attention (MHA) mechanism. We propose an alternative module that uses only a single shared projection matrix and multiple head embeddings (MHE), i.e. one per head. We empirically demonstrate that our MHE attention is substantially more memory efficient compared to alternative attention mechanisms while achieving high predictive performance retention ratio to vanilla MHA on several downstream tasks. MHE attention only requires a negligible fraction of additional parameters ($3nd$, where $n$ is the number of attention heads and $d$ the size of the head embeddings) compared to a single-head attention, while MHA requires $(3n^2 - 3n)d^2 - 3nd$ additional parameters.[1]

## 1 Introduction

Scaling pre-trained language models (PLMs) aims to enhance performance by increasing their size and capacity, leading to models with an unprecedented number of parameters (Kaplan et al., 2020; Chowdhery et al., 2022; Hoffmann et al., 2022). Just by increasing the size of PLMs and the pre-training data has yielded state-of-the-art performance on various natural language processing (NLP) tasks (Devlin et al., 2019; Liu et al., 2019; Clark et al., 2020; Raffel et al., 2020; Brown et al., 2020; Clark et al., 2022a; Ouyang et al., 2022; Touvron et al., 2023).

However, the pursuit of developing larger PLMs comes with large computational requirements. This has direct environmental implications such as large carbon emissions (Lacoste et al., 2019; Strubell et al., 2019; Weidinger et al., 2022), conflicting

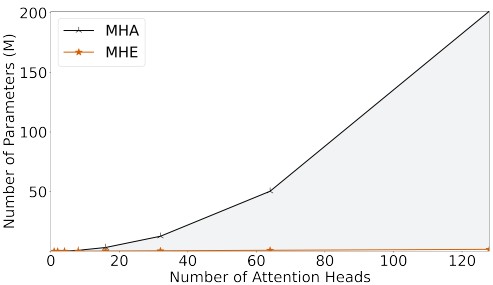

Figure 1: Number of parameters for an attention sublayer and different number of attention heads using multi-head attention MHA and our multi-head embedding attention MHE. We fix the dimension of attention to 64, only counting the parameters for projecting queries, keys, and values.

with the principles of Green artificial intelligence development (Schwartz et al., 2020). Moreover, scaling can hinder researchers with limited access to computing resources to participate in advancing the field (Schwartz et al., 2020). This results in inequalities, where only a privileged few can actively contribute, potentially impeding diversity and inclusivity (Weidinger et al., 2022).

The backbone of transformers (Vaswani et al., 2017) is the multi-head attention (MHA) module that extends the standard single-head attention (SHA) proposed by Cho et al. (2014). MHA applies an attention mechanism (i.e. head) multiple times for the same set of queries, keys and values by using a different set of parameters (i.e. projection matrices) for each of them. This results in MHA modules with a large memory footprint that increases with the number of layers and attention heads per layer in PLMs (Devlin et al., 2019; Brown et al., 2020; Ouyang et al., 2022; Touvron et al., 2023). Figure 1 shows how the number of parameters of a single attention sublayer increases with its number of attention heads.

Previous work has attempted to address this issue by proposing to share projection matrices or

---

[1] Code: https://github.com/HUIYINXUE/simpleMHE

eliminating them entirely to improve the parameter efficiency of MHA. Lan et al. (2020a) proposed sharing projection parameters for keys, queries and values across layers, while Kitaev et al. (2020) introduced a method for sharing the projection matrix between keys and values within each transformer layer. Additionally, similar approaches use a multi-query attention approach that uses a pair of global projection matrices for keys and values in each layer (Shazeer, 2019; Chowdhery et al., 2022; Ainslie et al., 2023). Furthermore, Yan et al. (2021) eliminate the projection matrices entirely and directly treat the input hidden states as both keys and values. In a different direction, Lee-Thorp et al. (2022) propose models that replace the attention blocks with token-mixture blocks (i.e. using linear or Fourier transformations) that contain fewer or no parameters compared to MHA.

Inspired by the position embeddings in transformers (Vaswani et al., 2017; Devlin et al., 2019), we aim to simplify and reduce the memory footprint of the MHA mechanism. We achieve this using only a single projection matrix for each of the keys, queries and values respectively shared across all attention heads, and one embedding per head (MHE).

Our contributions are as follows:

- We propose MHE, a novel attention module that uses shared projection matrices across heads that are modified by corresponding embedding heads. Our method generates multiple attention heads requiring only a small fraction of additional parameters compared to single-head attention.

- We empirically demonstrate that our MHE attention is substantially more parameter efficient compared to alternative attention mechanisms while achieving high predictive performance retention ratio (i.e. 92.9~98.7%) to MHA on several downstream tasks. MHE is $(3n^2 - 3n)d^2 - 3nd$ smaller than MHA for a single attention sublayer with $n$ attention heads and a hidden dimension of $d$ per head.

## 2 Related Work

### 2.1 Model Compression

To make PLMs memory efficient, previous work has focused on the following post-hoc model compression approaches (Ganesh et al., 2021; Tay et al., 2022).

**Quantization** Hubara et al. (2017) proposed representing weights using fewer bits to reduce memory requirements. Zadeh et al. (2020) introduced a method for identifying the outliers in weights and excluded them during quantization. Another direction involves additional training steps to adjust the quantized weights, i.e. quantization-aware training (Zafrir et al., 2019; Boo and Sung, 2020; Stock et al., 2020; Shen et al., 2020; Tambe et al., 2021; Tao et al., 2022). Bai et al. (2022) developed a more efficient post-training quantization approach that minimizes the reconstruction error incurred by quantization.

**Pruning** These compression approaches remove entirely parts of the network such as weights close to zero (Gordon et al., 2020; Mao et al., 2020; Chen et al., 2020) and weights that move towards zero during fine-tuning (Sanh et al., 2020; Tambe et al., 2021). Different to operating on individual weights, previous work attempted to remove structured blocks of weights or even architectural components such as attention heads and encoder layers (Fan et al., 2019; Prasanna et al., 2020; Khetan and Karnin, 2020; Li et al., 2020a; Lin et al., 2020; Tay et al., 2021).

**Knowledge Distillation** This set of techniques typically train a light-weight student model to mimic the outputs of a larger teacher PLM (Sun et al., 2019; Li et al., 2020b; Jiao et al., 2020; Sun et al., 2020; Li et al., 2021; Tahaei et al., 2022). In a similar direction, smaller PLMs have been recently fine-tuned on text generated by larger PLMs (Chiang et al., 2023; Taori et al., 2023).

**Weight Matrix Decomposition** Previous work also proposed replacing large weight matrices by the product of two smaller ones for reducing model size and runtime memory. Weight matrix decomposition has been applied to linear layers (Mao et al., 2020; Ben Noach and Goldberg, 2020), the embedding matrix (Lan et al., 2020b; Tambe et al., 2021; Wang et al., 2022), and attention blocks (Hu et al., 2022; Wang et al., 2022).

**Embedding Matrix Compression** Finally, various attempts have been introduced for compressing the embedding matrix during pre-training and fine-tuning (Xue et al., 2022; Clark et al., 2022b; Xue and Aletras, 2022).

## 2.2 Improving Attention Efficiency

Previous work on making attention more efficient includes efforts towards (1) speeding-up pairwise computations between token representations; and (2) parameter efficiency.

**Computational Efficiency** While improving computational efficiency of attention is out of the scope of our paper, we provide a brief overview of previous work since it is complementary to parameter efficiency. One approach to speed up attention computation is by reducing the number of similarity computations between representations in different positions using predefined local windows, fixed or dynamic strides (Child et al., 2019; Zaheer et al., 2020; Beltagy et al., 2020; Kitaev et al., 2020). Other methods leverage the approximation of SoftMax to change the order of matrix multiplications, resulting in lower computational complexity (Katharopoulos et al., 2020; Choromanski et al., 2021; Schlag et al., 2021; Qin et al., 2022). Related approaches along this direction proposed kernel functions that require additional parameters (Choromanski et al., 2021; Wang et al., 2020). Finally, Dao et al. (2022) proposed improvements in GPU memory access to optimize and accelerate the MHA computation.

**Memory Efficiency** Lan et al. (2020a) introduced a method for sharing the projection parameters for queries, keys and values across transformer layers. Furthermore, Kitaev et al. (2020) proposed sharing the projection matrix between keys and values within each layer. Additionally, other methods use a multi-query attention approach that shares projection weights for keys and values across different heads (Shazeer, 2019; Chowdhery et al., 2022; Ainslie et al., 2023), while Yan et al. (2021) directly treat the input hidden states as both keys and values. In a different direction, Lee-Thorp et al. (2022) proposed replacing the attention blocks with faster token-mixture blocks consisting of a few parameters or no parameters at all. This includes methods such as linear or Fourier transformations in the token-mixture block. However, these approaches tend to yield lower predictive performance compared to MHA.

## 3 Multiple Head Embeddings Attention

Inspired by the absolute position embeddings (Vaswani et al., 2017; Devlin et al., 2019) for distinguishing the representation of the same token in different contexts, we propose Multiple Head Embeddings (MHE) attention. MHE uses a shared 'seed' projection matrix that is subsequently combined with distinct head embeddings to generate multiple attention heads.

### 3.1 Multi-head Attention (MHA)

We first begin by formally defining MHA. MHA consists of different projection matrices ($\mathbf{W}_i^Q, \mathbf{W}_i^K, \mathbf{W}_i^V \in \mathbb{R}^{d_m \times d_h}, i = 1, ..., n$, where $d_m$ is the dimension of the input representation and $d_h$ is the dimension of $n$ attention heads) for queries ($Q$), keys ($K$) and values ($V$) per head, $3 \times n$ in total. It is computed as follows:

$$\mathbf{Q}_i, \mathbf{K}_i, \mathbf{V}_i = \mathbf{X}\mathbf{W}_i^{Q,K,V} \tag{1}$$

$$\mathbf{H}_i = \text{Att}(\mathbf{Q}_i, \mathbf{K}_i, \mathbf{V}_i) \tag{2}$$

$$= \text{SoftMax}\left(\frac{\mathbf{Q}_i\mathbf{K}_i^\top}{\sqrt{d_h}}\right)\mathbf{V}_i \tag{3}$$

Note that we use scale-dot attention, but our method can be used with any other attention mechanism.

### 3.2 Seed Projection Matrix

Unlike MHA that uses different projection matrices per head, MHE attention employs only a single projection matrix for each of the queries, keys and values, $\mathbf{W}^Q, \mathbf{W}^K, \mathbf{W}^V \in \mathbb{R}^{d_m \times d_h}$. These matrices are shared across all attention heads.

We obtain query, key and values projections of the input sequence $\mathbf{X}$ as follows:

$$\mathbf{Q}, \mathbf{K}, \mathbf{V} = \mathbf{X}\mathbf{W}^{Q,K,V} \tag{4}$$

### 3.3 Attention Head Embeddings

Using a seed projection matrix for $\mathbf{Q}, \mathbf{K}, \mathbf{V}$ is equivalent to a single-head attention (SHA) module. Therefore, we need a mechanism to transform the seed projection matrices to obtain different attention head. For this purpose, we represent each attention head $i$ by specific head embeddings $\mathbf{e}_i^Q, \mathbf{e}_i^K, \mathbf{e}_i^V \in \mathbb{R}^{d_h}, i = 1, ..., n$ for queries, key and values. These embeddings have a substantially smaller memory footprint compared to using different projection matrices per head. The contextualized representation $\mathbf{H}_i$ of the entire input sequence $\mathbf{X}$ for head $i$ is computed as follows:

$$\tilde{\mathbf{Q}}_i, \tilde{\mathbf{K}}_i, \tilde{\mathbf{V}}_i = \psi(\mathbf{Q}; \mathbf{K}; \mathbf{V}, \mathbf{e}_i^{Q,K,V}) \tag{5}$$

$$\mathbf{H}_i = \text{Att}(\tilde{\mathbf{Q}}_i, \tilde{\mathbf{K}}_i, \tilde{\mathbf{V}}_i) \tag{6}$$

where $\psi(\cdot)$ is a function that modifies the query, key and value matrices with a corresponding head embedding $\mathbf{e}_i$.

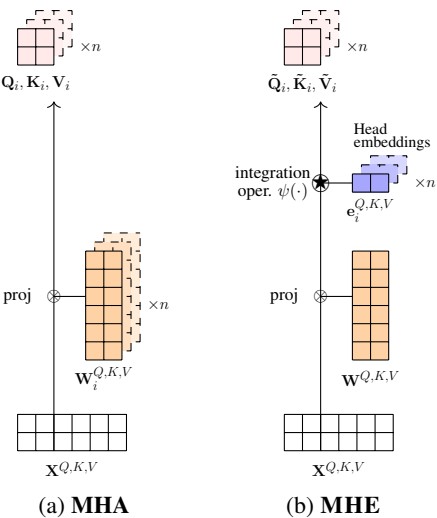

(a) **MHA**  (b) **MHE**

Figure 2: Multi-head attention (left) requires $3 \times n$ projection matrices for queries, keys and values ($\mathbf{W}^{Q,K,V}$) where $n$ is the number of attention heads. Multi-head embedding attention (right) uses only three projection matrices and $3 \times n$ head embeddings.

### 3.4 Modifying Queries, Keys and Values with Head Embeddings

We propose two MHE variants, one adds and the other multiplies the head embeddings with the seed projection matrices.

**MHE-ADD:** Motivated by the absolute position embedding (Devlin et al., 2019), we use the addition operation in Equation 5, represented as $\psi(\mathbf{A}, \mathbf{b}) := \mathbf{A} + \mathbf{b}$, where $\mathbf{A} \in \{\mathbf{Q}, \mathbf{K}, \mathbf{V}\}$ and $\mathbf{b} \in \{\mathbf{e}^Q, \mathbf{e}^K, \mathbf{e}^V\}$ respectively.

**MHE-MUL:** Likewise, motivated by the rotary position embedding (Su et al., 2021), MHE-MUL employs multiplication as the integrating operation in Equation 5 as $\psi(\mathbf{A}, \mathbf{b}) := \mathbf{A} \odot (\mathbf{b} + 1)$, where $\odot$ represents the Hadamard product.[2]

Figure 2 shows an overview of the MHE mechanism compared to MHA.

## 4 Experimental Setup

### 4.1 Attention Mechanisms

We compare our MHE attention with the following attention mechanisms:[3]

---

[2]We add 1 to avoid elements in queries, keys and values become too small during initialization.

[3]We have also experimented with Linear and Fourier token-mixture models (Lee-Thorp et al., 2022) yielding substantially lower performance. For full results of these methods, see Appendix A.

- **Multi-head Attention (MHA):** This is the original multi-head attention mechanism (Vaswani et al., 2017; Devlin et al., 2019).

- **Single-head Attention (SHA):** Similar to MHA but using only one attention head.

- **EL-ATT:** Introduced by Yan et al. (2021), this attention variant completely eliminates the projection matrices for all keys and values.

- **MQA:** Introduced by Shazeer (2019), this approach uses shared projection matrices for keys and values across all attention heads. Note that different projection matrices are used for queries across heads.

- **SKV:** Introduced by Kitaev et al. (2020), this attention variant enforces keys and values to share the same projection matrix within each attention module.

### 4.2 Data

We experiment with a diverse range of tasks including: (1) two standard natural language understanding benchmarks in English, GLUE (Wang et al., 2018) and SUPERGLUE (Wang et al., 2019); (2) two question and answering benchmarks in English, SQUAD V1.1 (Rajpurkar et al., 2016) and SQUAD V2.0 (Rajpurkar et al., 2018); (3) WMT-14 English-to-German machine translation (Bojar et al., 2014); and (4) two language modelling datasets in English WIKITEXT-103 (Merity et al., 2017) and PENN TREEBANK (Marcus et al., 1993).

### 4.3 Models

We test all different attention variants on two architectures: (1) encoder-only transformer (Devlin et al., 2019) and (2) encoder-decoder transformer (Vaswani et al., 2017).

**Encoder-only** For GLUE, SUPERGLUE, SQUAD V1.1 and SQUAD V2.0, we use a BERT-base architecture. This consists of 12 transformer layers, embedding size of 768, hidden states dimension of 768, 12 attention heads and a maximum sequence length of 512.

**Decoder-only** We also test a decoder-only model using the GPT2-base architecture on WIKITEXT-103, PENN TREEBANK and GLUE. GPT2-base consists of 12 transformer layers, embedding size of 768, hidden states dimension of 768, 12 attention heads and a maximum sequence length of 512.

**Encoder-decoder** For WMT-14, we train an encoder-decoder transformer from scratch. It consists of 12 layers (6 for the encoder and decoder respectively), an embedding size of 512, hidden states dimension of 512 and 8 attention-heads and a maximum sequence length of 100.

We set the number of attention heads to 1 for all SHA models. Experimenting with larger models and different number of attention heads is out of the scope of our paper and left for future work due to limited access to computing resources.

### 4.4 Implementation Details

**Pre-training** We pre-train all models on the English Wikipedia and BookCorpus (Zhu et al., 2015) from HuggingFace (Lhoest et al., 2021) for up to 1M steps with a batch size of 128. We choose masked language modelling as the pre-training objective. For all models, we use a 30K WordPiece vocabulary (Devlin et al., 2019).

**Fine-tuning and Training** For GLUE, SUPER-GLUE, SQUAD V1.1 and SQUAD V2.0, we fine-tune all pre-trained models up to 20 epochs with early stopping fixing the batch size to 32. For each task, we use five different seeds and report the average.

We train the encoder-decoder model from scratch on the training set of WMT-14 English-to-German machine translation dataset up to 100K steps with a batch size of 256. WMT-14 contains 4.5M sentence pairs and evaluate on its test set. We train the tokenizer using byte-pair-encoding (Sennrich et al., 2016) with 37K merging steps on the training set. We enable both source language and target language to share the vocabulary. We use one random seed and report the average on the last five epochs. We optimize all models using AdamW (Loshchilov and Hutter, 2019).

**Hyperparameters** Hyperparameter selection details are in Appendix B.

**Hardware** For pre-training, we use four NVIDIA Tesla A100 GPUs and one for fine-tuning on downstream tasks.

### 4.5 Predictive Performance Evaluation

For GLUE, SUPERGLUE, SQUAD V1.1 and SQUAD V2.0, we use the official metric of each task (see Appendix A for details on metrics for each task). We report F1 score for SQUAD V1.1

and SQUAD V2.0. We use BLEU to report performance in WMT-14 English-to-German machine translation task. We use perplexity (PPL) to report generative performance on WIKITEXT-103 and PENN TREEBANK by fixing the stride length to 256.

### 4.6 Memory Efficiency Evaluation

Furthermore, we use the following metrics to measure and compare the memory efficiency of MHE and the baselines.

- **Performance Retention Ratio:** We compute the ratio between the predictive performance of each attention mechanism compared to MHA upper-bound baseline performance (the higher the better).
  For direct indicator (e.g. accuracy etc.):

  $$\text{PRR} = \frac{\text{score}_{\text{model}}}{\text{score}_{\text{MHA}}}$$

  For inverse indicator (e.g. perplexity etc.):

  $$\text{PRR} = 1 - \frac{\text{score}_{\text{model}} - \text{score}_{\text{MHA}}}{\text{score}_{\text{MHA}}}$$

- **Performance Elasticity of Parameters:** Inspired by the concept of elasticity in economics (Bittermann, 1934), which measures the responsiveness of an economic variable (e.g. investment demand) to a change in another (e.g. interest rate), we extend it to measure the parameter utilization rate of a target model compared to the SHA lower-bound. The performance elasticity of parameters (PEoP) indicates how effectively parameters contribute to predictive performance, compared to SHA. It is computed as follows:
  For direct indicator (e.g. accuracy etc.):

  $$\text{PEoP} = \frac{(\text{score}_{\text{model}}/\text{score}_{\text{SHA}}) - 1}{(\text{params}_{\text{model}}/\text{params}_{\text{SHA}}) - 1}$$

  For inverse indicator (e.g. perplexity etc.):

  $$\text{PEoP} = -\frac{(\text{score}_{\text{model}}/\text{score}_{\text{SHA}}) - 1}{(\text{params}_{\text{model}}/\text{params}_{\text{SHA}}) - 1}$$

  PEoP quantifies the extent to which a model's performance can be boosted with 1% additional parameters compared to a baseline model (the higher the better).[4]

---

[4]We subtract 1 in both nominator and denominator, following the original definition of elasticity.

| Attention | #params | GLUE | | | SUPERGLUE | | | SQUAD v1.1 | | | SQUAD v2.0 | | |
|---|---|---|---|---|---|---|---|---|---|---|---|---|---|
| | | Acc | PRR | PEoP | Acc | PRR | PEoP | Acc | PRR | PEoP | Acc | PRR | PEoP |
| SHA | 8.85M | 79.2 | 96.7 | - | 67.1 | 95.1 | - | 82.5 | 93.1 | - | 67.6 | 87.8 | - |
| MHA | 28.32M | 81.9 | 100.0 | 0.02 | 70.5 | 100.0 | 0.02 | 88.6 | 100.0 | 0.03 | 77.0 | 100.0 | 0.06 |
| EL-ATT | 14.16M | 80.3 | 98.0 | 0.02 | 69.5 | 98.5 | 0.06 | 86.5 | 97.6 | 0.08 | 72.2 | 93.8 | 0.11 |
| MQA | 15.34M | 81.3 | 99.2 | 0.04 | 69.3 | 98.2 | 0.04 | 86.7 | 97.9 | 0.07 | 74.8 | 97.1 | 0.15 |
| SKV | 21.23M | **81.4** | **99.4** | 0.02 | **69.9** | **99.1** | 0.03 | **88.1** | **99.4** | 0.05 | **75.9** | **98.6** | 0.09 |
| MHE-ADD | 8.88M | 80.4 | 98.2 | 4.92 | 69.1 | 97.9 | 9.44 | 83.7 | 94.5 | 4.65 | 71.8 | 93.2 | 19.88 |
| MHE-MUL | 8.88M | 80.6 | 98.3 | **5.53** | 69.6 | 98.7 | **12.07** | 85.9 | 97.0 | **13.19** | 72.3 | 93.9 | **22.25** |

Table 1: Results of the encoder-only architecture on GLUE, SUPERGLUE, SQUAD V1.1 and SQUAD V2.0 dev sets with performance retention ratio (PRR) and performance elasticity of parameters (PEoP) over five runs. **Bold** values denote best performing method in each benchmark.

| Attention | #params | GLUE | | | WIKITEXT-103 | | | PENN TREEBANK | | |
|---|---|---|---|---|---|---|---|---|---|---|
| | | Acc | PRR | PEoP | PPL | PRR | PEoP | PPL | PRR | PEoP |
| SHA | 8.85M | 75.3 | 97.2 | - | 62.0 | 55.8 | - | 68.1 | 46.3 | - |
| MHA | 28.32M | 77.5 | 100.0 | 0.01 | 43.0 | 100.0 | 0.14 | 44.3 | 100.0 | 0.16 |
| EL-ATT | 14.16M | 76.6 | 98.9 | 0.03 | 57.1 | 67.2 | 0.13 | 56.1 | 73.4 | 0.29 |
| MQA | 15.34M | 76.9 | 99.2 | 0.03 | 49.7 | 84.4 | 0.27 | 49.3 | 88.7 | 0.38 |
| SKV | 21.23M | **77.1** | **99.5** | 0.02 | **46.2** | **92.6** | 0.18 | **45.5** | **97.3** | 0.24 |
| MHE-ADD | 8.88M | 75.8 | 97.8 | 2.18 | 54.0 | 74.4 | 41.29 | 55.3 | 75.2 | 60.15 |
| MHE-MUL | 8.88M | 76.7 | 99.0 | **5.92** | 53.8 | 74.9 | **42.32** | 50.7 | 85.6 | **81.76** |

Table 2: Results of decoder-only architecture on GLUE dev sets and WIKITEXT-103, PENN TREEBANK test sets with performance retention ratio (PRR) and performance elasticity of parameters (PEoP) over five runs. **Bold** values denote best performing method in each benchmark.

## 5 Results

### 5.1 Predictive Performance Comparison

Table 1 presents results on GLUE, SUPERGLUE, SQUAD V1.1 and SQUAD V2.0 for our MHE variants and all baselines. We first observe that both the performance of our MHE-ADD and MHE-MUL are comparable to the vanilla MHA on two text classification benchmarks (80.4, 80.6 vs. 81.9 on average GLUE and 69.1, 69.6 vs. 70.5 on average SUPERGLUE) with high performance retention ratios (PRR) between 97.9% and 98.7%. On question answering tasks SQUAD V1.1 and SQUAD V2.0, both MHE variants are also competitive, with PRRs higher than 93%.

Similar results are observed on the WMT-14 English-to-German machine translation task for the encoder-decoder transformer. According to Table 3, MHE-ADD and MHE-MUL achieve BLEU scores of 23.0 and 23.6, respectively. The performance of MHE-MUL is negligibly lower than that of MHA (24.8) while being substantially smaller.

Consistent results for the decoder-only transformer are shown in Table 2. The PRRs for MHE-ADD and MHE-MUL on GLUE are still high (i.e. 97.8% and 99.0%). While using the intrinsic met-

rics for evaluation, MHE-MUL leads to the perplexities of 53.8 and 50.7 compared to 43.0 and 44.3 for MHA on WIKITEXT-103 and PENN TREEBANK respectively, indicating a stable PRR higher than 74.9%.

In all tasks, MHE consistently outperforms SHA by a large margin with only 0.03M extra parameters, i.e. 0.6~17.4. For example, 69.6 vs. 67.1 in SUPERGLUE, 72.3 vs. 67.6 in SQUAD V2.0, 23.6 vs. 22.5 in WMT-14 and 62.0 vs. 53.8 in WIKITEXT-103 for the MHE-MUL variant. We also note that MQA and SKV attention mechanisms generally perform better than MHE, however they are 1.7 and 2.4 times larger than MHE, i.e. 15.34M and 21.23M vs. 8.88M parameters. It is worth noting that MHE-MUL outperforms EL-ATT on three out of five benchmarks, despite having nearly half the parameters in the attention module.

### 5.2 Memory Efficiency Comparison

Our results so far indicate that performance increases with the number of attention mechanism parameters, which is expected. Next, we inspect how efficiently different attention mechanisms uti-

lize their parameters [5]. Tables 1 and 3 show how parameter efficient our two MHE attention variants and all baselines are, measured in PEoP. Note that PEoP scores for SHA cannot be computed as it is used as the point for reference model. We also report PRR using MHA as a baseline for completeness, however this metric does not take the model size into account.

We first observe in Table 1 that both our MHE-ADD and MHE-MUL achieve the highest PEoP scores on the two natural language understanding benchmarks (4.92, 5.53 on GLUE, and 9.44, 12.07 on SUPERGLUE) and two question answering tasks (4.65, 13.19on SQUAD V1.1, and 19.88, 22.25 on SQUAD V2.0). In contrast, vanilla MHA results in the lowest PEoP score among all models as expected, ranging from 0.02 to 0.06. It indicates the memory inefficiency of MHA.

The PEoPs of more light-weight EL-ATT and SKV are similar to that of MHA (0.02) on average GLUE, barely 4 ‰of that of MHE, indicating they are far more memory-inefficient compared to MHE.

Similar findings are observed in WMT-14 for the encoder-decoder models depicted in Table 3. MHE-ADD and MHE-MUL achieve PEoP scores of 20.0 and 27.9, respectively. In contrast, the PEoP scores of MHA, EL-ATT MQA and SKV are close to zero (barely 0.1). This means that investing more parameters into their attention modules would not bring proportional benefits in predictive performance. Even for the SKV which is half the size of MHA and achieves high PRR, when the number of parameters increase by 1%, the BLEU score increases a negligible 0.1%, while evolving from SHA. However, with the same number of parameters, our most memory-inefficient MHE-MUL is able to improve the BLEU score by 11.0%. Such rate of return is 110 times larger than that of SKV. Leveraging the head embeddings by adding only a negligible number of parameters efficiently improves the predictive performance.

We further observe that MHE-ADD and MHE-MUL are architecture-agnostic, obtaining similar memory efficiency for the decoder-only model in Table 2. Both our MHE-ADD and MHE-MUL achieve the highest PEoP scores on the two language modelling benchmarks (41.29, 42.32 on WIKITEXT-103 and 60.15 and 81.76 on PENN

---

[5]For a detailed report on the memory usage of different attention mechanisms, see Appendix C.

| Attention | #params | BLEU | PRR | PEoP |
|---|---|---|---|---|
| SHA | 6.49M | 22.5 | 90.8 | - |
| MHA | 18.87M | 24.8 | 100.0 | 0.1 |
| EL-ATT | 9.44M | 23.9 | 96.6 | 0.1 |
| MQA | 10.62M | 24.2 | 97.6 | 0.1 |
| SKV | 14.16M | **24.7** | **99.5** | 0.1 |
| MHE-ADD | 6.52M | 23.0 | 92.9 | 5.5 |
| MHE-MUL | 6.52M | 23.6 | 95.0 | **11.0** |

Table 3: BLEU scores on WMT-14 English to German machine translation task with performance retention ratio (PRR) and performance elasticity of parameters (PEoP). **Bold** values denote best performing method in each benchmark.

TREEBANK) and GLUE (2.18 and 5.92). At the same time, MHA fail to perform well on GLUE and PENN TREEBANK with a PEoP of 0.01 and 0.16 respectively. MHE-ADD and MHE-MUL also consistently outperform other efficient-attention variants (i.e. EL-ATT, MQA and SKV) by 72~340 times on PEoP across the three benchmarks.

In all tasks, MHE consistently outperforms MHA by orders of magnitude in parameter efficiency. We also note that EL-ATT, MQA and SKV only lead to PEoP scores with the same magnitude as MHA. This highlights the more superior parameter utilization of MHE attention variants, achieving state-of-the-art memory-efficiency.

## 5.3 Theoretical Memory Complexity

Table 4 presents the theoretical memory complexity and the total number of parameters of our two MHE and baseline attention mechanisms in a single transformer sublayer. First, we see that the theoretical memory complexity of MHA and other efficient parameters (EL-ATT, MQA and SKV) are quadratic with the number of attention heads, while our MHE are the only two variants having the complexity linear with the attention heads similar to SHA.

Taking a closer look at the rightmost column in Table 4, we observe that the number of extra parameters of all attention variants compared to SHA have a quadratic relationship to both the number $n$ and the dimension of attention heads $d$, except our two MHE variants. MHE only requires a relatively small fraction of additional parameters compared to SHA.

| Attention | Complexity | #Params | #Params (+) |
|---|---|---|---|
| SHA | $\mathcal{O}(n)$ | $3d^2n$ | 0 |
| MHA | $\mathcal{O}(n^2)$ | $3d^2n^2$ | $(3n^2 - 3n)d^2$ |
| EL-ATT | $\mathcal{O}(n^2)$ | $d^2n^2$ | $(n^2 - 3n)d^2$ |
| MQA | $\mathcal{O}(n^2)$ | $d^2n^2 + 2d^2n$ | $(n^2 - n)d^2$ |
| SKV | $\mathcal{O}(n^2)$ | $2d^2n^2$ | $(2n^2 - 3n)d^2$ |
| MHE (ours) | | | |
| -ADD | $\mathcal{O}(n)$ | $3d^2n + 3dn$ | $3nd$ |
| -MUL | $\mathcal{O}(n)$ | $3d^2n + 3dn$ | $3nd$ |

Table 4: Memory complexity regarding the number of parameters in each attention sublayer, while fixing the dimension of attention heads to $d$. $n$ denotes the number of attention heads. To simplify, the dimension of hidden states $d_m$ is set to $nd$. The last projection for pooling attention heads is excluded.

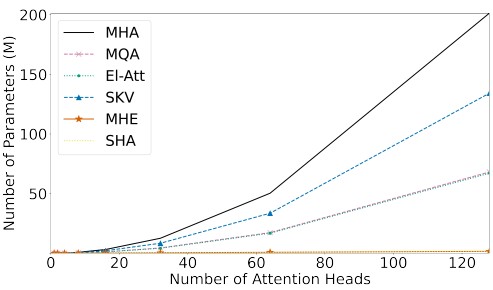

Figure 3: Number of parameters per attention sublayer, while scaling the number of attention heads in different attention variants. We fix the dimension of attention to 64.

## 5.4 Scaling the Number of Attention Parameters

Delving deeper to the effect of scaling to memory footprint, we show in Figure 3 the total number of parameters needed for a single attention module (e.g. in an encoder layer). We fix the dimension of attention heads to 64 commonly used by BERT (Devlin et al., 2019), RoBERTa (Liu et al., 2019), GPT-2 (Radford et al., 2019), BART (Lewis et al., 2020) and T5 (Raffel et al., 2020). In general, we note that the number of parameters in MHA could reach more than 200M if employing 128 attention heads. At the same time, SKV, MQA and EL-ATT would require 2/3, 1/3 and 1/3 of that number respectively. In contrast, MHE only accounts for 1% of the MHA parameters.

Moreover, we also present in Figure 4 the total number of parameters required across attention variants when stacking 12, 24 and 48 layers along with 32 and 64 attention heads respectively. We also fix the dimension of attention heads to 64. We can observe, when the number of attention head reaches 64, MHA with 24 layers already occupies more than 1B parameters, while EL-ATT and MQA reach 0.8B parameters with 48 layers. SKV takes 24 layers to reach 0.8B parameters. However, the total number of parameters in MHE attention does not exceed 0.1B even when scaling to 48 layers with 64 attention heads. It is also clear that scaling the attention module to 48 layers, 32 attention heads and 12 layers needs a comparable number of parameters for MHA, EL-ATT, MQA or SKV. This indicates, that LLM developers have to make a choice whether doubling the number of attention

heads or cutting down the number of layers to a quarter when working under a tight memory budget. However, MHE does not suffer by such issues.

Further, we project these estimates to the popular GPT-3 model (Brown et al., 2020). It is a decoder-only model with 96 decoder layers, 96 attention heads per layer, and a head dimension of 128. The vanilla multi-head attention module requires a massive 43.48B parameters. However, using MHE attention, this number can be significantly reduced to 0.46B parameters, i.e. approximately a reduction by 98.9%.[6] Comparing this to other parameter-efficient attention variants such as EL-ATT (14.50B parameters), MQA attention (14.80B parameters), and SKV attention (28.99B parameters), it becomes evident that our MHE offers better memory efficiency. This makes it a compelling alternative for memory-constrained scenarios. See Appendix D for a detailed study on the robustness of MHE to model size changes (i.e. scaling).

## 6 Discussion

MHA enables the model to attend to information from different representation subspaces at different positions (Vaswani et al., 2017). It uses distinct projection matrices for each attention head and integrates the information from these different representation subspaces. However, Vaswani et al. (2017) did not explore different methods for performing space transformations per head.

Previous work has pointed out that overparameterized models might have a low intrinsic dimension. Therefore, transforming the projection

---

[6]It would have been great to report results by pre-training our own MHE GPT-3 model, however this is prohibitive with the modest compute we have available.

matrices to smaller low-rank ones usually does not severely harm model predictive performance (Li et al., 2018; Aghajanyan et al., 2020). Meanwhile, the classic MHA approach also does not impose any constraints on the orthogonality of these subspaces during pre-training and fine-tuning. The column vectors in those projection matrices could be highly collinear, i.e. the projection matrices could be rank-deficient. As a result, its inner-working mechanism could be simply understood as introducing levels of variation to the encoded representation of the same token at the same position across different heads.

Our MHE approach is possible to achieve memory efficiency (similar to SHA) together with high PRR compared to MHA by mimicking the position embeddings for representing different attention heads.

On one hand, the addition operation in MHE-ADD is used for transforming the keys, queries and values. This can be seen as a small distortion of the subspace obtained through projection, followed by rotation. For an input representation, the difference between the projected and injected (i.e. through head embedding addition) queries, keys and values is a constant vector across any pair of heads. On the other hand, the MHE-MUL approach employs a multiplication operation, which more aggressively distorts and reshapes the keys, queries and values subspaces. Head embeddings in MHE-MUL play a role as the scaling factors, respectively stretching each dimension of the input representation. Thus, the difference between the keys, queries, and values generated by different heads for the same input representation, is a vector parallel to the projected input. This vector is dependent on the specific input, unlike the constant vector in MHE-ADD.

Interestingly, our experimental results consistently show that the multiplication operation outperforms addition in the majority of benchmarks. This corroborates findings of a previous empirical study by Su et al. (2021) that compared rotary position embeddings (somehow analogous to MHE-MUL) with absolute position embeddings (analogous to MHE-ADD).

## 7 Conclusions

We have proposed MHE attention that employs a single shared projection matrix along with multiple head embeddings, to simplify and reduce the memory footprint of the MHA. Our experi-

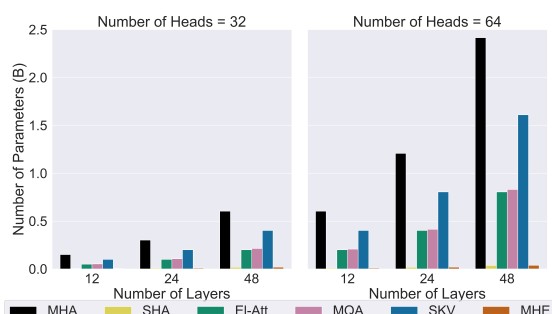

Figure 4: Total number of parameters in attention sub-layers, while scaling the number of attention layers to 12, 24 and 48 with 32 attention heads and 64 attention heads respectively. We fix the dimension of attention to 64.

mental results have demonstrated that MHE attention exhibits superior memory efficiency compared to other memory-efficient attention variants, while achieving high predictive performance ratio to MHA on various downstream tasks. Compared to a single-head attention, MHA requires $(3n^2 - 3n)d^2$ parameters for $n$ attention heads and head dimensionality $d$, while MHE barely requires a negligible $3nd$. For future research, we plan to investigate scaling up MHE models and explore its linguistic capabilities (Vulić et al., 2020; Koto et al., 2021).

## Limitations

We experiment only using 'base' size models without experimenting with larger architectures, due to limited access to computational resources. Similarly, we did not experiment with decoder only architectures (Brown et al., 2020) which we leave for future work. We have not combined our MHE method with computationally efficient attention methods with linear complexity, such as Linformer (Wang et al., 2020). We expect that it would speed up computation of MHE, but it is out of the scope of our paper.

## Acknowledgments

We would like to thank Constantinos Karouzos, Miles Williams and the anonymous reviewers for their invaluable feedback.

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

# A Reported Metrics for Each Task

We evaluate all models on GLUE (Wang et al., 2018), SUPERGLUE (Wang et al., 2019), SQUAD v1.1 (Rajpurkar et al., 2016) and SQUAD v2.0 (Rajpurkar et al., 2018). We report matched accuracy for MNLI, Matthews correlation for CoLA, Spearman correlation for STS, F1 score for QQP, CB, MultiRC and SQUAD and accuracy for all other tasks. Table 5 and Table 6 present results on GLUE and SUPERGLUE respectively for our MHE-FORMERS models and all baselines with the encoder-only architecture. Table 7 and 8 present results of the scores and performance elasticity of parameters (PEoP) across all models over each task in GLUE and SUPERGLUE. Table 10 presents results on GLUE for our MHE-FORMERS models and all baselines with the decoder-only architecture. Table 10 presents results of the scores and performance elasticity of parameters (PEoP) across all models over each task in GLUE.

# B Hyperparameters

The hyperparameters used in pre-training are listed in Table 11. The hyperparameters used in fine-tuning are listed in Table 12.

# C Memory Usage

To further illustrate the memory-efficiency of our MHE models compared to the baselines, we take the BERT-base architecture (12 attention heads, each with a dimension of 64) as an example, and measure the memory usage per attention block as in Section 2.1.1 from Smith et al. (2022) and report the memory usage saving ratio (%) during the attention calculation in Table 13:

The calculation is based on inputs with batch size of 32, hidden dimension of 768, sequence length of 512 and fp16 mixture precision training using the following formula:

- Memory(weights)=#params*(2+4) bytes;

- Memory(gradients)=#params*(2+4) bytes;

- Memory(Adam states)=#params*(4+4) bytes;

- Memory(activations)= batch-size*sequence-length*hidden-dimension*2 bytes.

From Table 13, we observe the memory usage saving ratio of our proposed MHE is 2.75 times better than SKV, 1.50 times better than MQA and 1.37 times better than EL-ATT, which indicates a SotA memory saving capabilities compared to all other parameter-efficient attention variants.

# D Robustness to Scaling

We also conduct experiments to observe the effectiveness and the robustness of our best MHE-MUL while scaling the model size.

Table 14 presents average accuracy on two text classification benchmarks (GLUE and SUPERGLUE), perplexities on two language modelling benchmarks (WIKITEXT-103 and PENN TREEBANK) with their corresponding performance retention ratio (PRR) for MHA and MHE-MUL in both encoder-only and decoder-only architecture across different model sizes.[7] For the encoder-only models, we observe that the PRR of MHE-MUL remains stable on GLUE (from 98.4% to 98.7%) and SUPERGLUE (from 98.7% to 96.2%) while scaling the number of parameters in the attention blocks to 3.5 times larger. For the decoder-only models, the PRR on GLUE for MHE-MUL stabilizes at 97.9% (i.e. 1.1% lower) after scaling. Surprisingly, the PRR of MHE-MUL increases on WIKITEXT-103 (from 74.9% to 95.2%) and PENN TREEBANK (from 85.6% to 88.5%) while scaling to MEDIUM size.

Similar results are observed for the encoder-decoder architecture on WMT14 machine translation task. According to Table 15, we first observe the PRR of MHE-MUL remains stable (i.e. between 91.5% and 96.0%) across all different sizes, where the number of parameters in the corresponding MHA ranges from 19.87M to 75.50M. Meanwhile, we also observe that making the model deeper by stacking more encoder and decoder layers results in a steady increment on PRR for MHE-MUL (e.g. 93.6%, 95.0% and 96.0% respectively, for 8 layers, 12 layers and 16 layers in total). Moreover, for the same number of parameters in the attention, wider attention heads consistently leads to a better PRR for MHE-MUL, i.e. 91.5%, 95.0% and 95.3% for the dimensionality of 32, 64 and 128 of attention heads respectively.

These results indicate MHE consistently achieves good performance retention ratios and is robust to model size change.

---

[7]BASE: 12 encoder/decoder layers, each containing 12 attention heads; LARGE/MEDIUM: 24 encoder/decoder layers, each containing 16 attention heads.

| ATTENTION | MNLI | QNLI | QQP | RTE | SST | MRPC | CoLA | STS | GLUE Avg. |
|---|---|---|---|---|---|---|---|---|---|
| SHA | 80.5(0.3) | 87.5(0.2) | 86.7(0.1) | 63.6(0.9) | 90.7(0.3) | 85.1(0.7) | 53.8(1.1) | 85.8(0.4) | 79.2(0.1) |
| MHA | 83.4(0.1) | 89.8(0.3) | 87.8(0.1) | 67.6(1.5) | 92.0(0.3) | 86.8(0.4) | 59.6(1.3) | 88.5(0.3) | 81.9(0.3) |
| EL-ATT | 81.7(0.1) | 88.4(0.2) | 87.3(0.2) | 67.6(1.0) | **91.7**(0.6) | 85.9(0.7) | 52.4(1.7) | 87.7(0.2) | 80.3(0.3) |
| MQA | **82.6**(0.1) | 88.8(0.2) | 87.3(0.1) | 66.5(0.9) | 91.4(0.5) | 87.3(0.2) | **58.4**(1.3) | 87.9(0.2) | 81.3(0.2) |
| SKV | **82.6**(0.1) | **89.4**(0.3) | **87.7**(0.1) | **68.2**(1.7) | 91.6(0.3) | **87.4**(0.6) | 56.2(1.2) | **88.6**(0.2) | **81.4**(0.2) |
| FNET | 76.3(0.1) | 83.8(0.1) | 84.8(0.1) | 63.2(2.0) | 88.4(0.7) | 78.0(0.4) | 43.2(2.5) | 83.7(0.3) | 75.2(0.6) |
| LINEAR | 75.4(0.1) | 81.4(0.1) | 85.5(0.2) | 54.7(2.3) | 90.4(0.4) | 72.2(0.6) | 50.3(1.0) | 70.9(0.5) | 72.6(1.1) |
| MHE-ADD | 81.5(0.2) | 87.8(0.2) | 87.2(0.1) | 66.9(2.0) | 90.5(0.4) | 87.2(0.3) | 54.7(0.9) | 87.7(0.1) | 80.4(0.2) |
| MHE-MUL | 81.9(0.1) | 87.9(0.1) | 87.4(0.1) | 67.1(1.5) | 91.1(0.5) | 85.4(0.5) | 56.6(1.7) | 87.3(0.2) | 80.6(0.2) |
| MHA(M) | 84.4(0.2) | 91.1(0.4) | 84.0(0.6) | 70.5(1.0) | 92.0(0.2) | 87.2(0.8) | 62.5(1.0) | 88.8(0.2) | 82.6(0.4) |
| MHE-MUL (M) | 82.7(0.2) | 89.2(0.4) | 87.2(0.2) | 67.9(0.4) | 90.7(0.3) | 86.3(1.0) | 59.8(1.8) | 88.0(0.2) | 81.5(0.3) |

Table 5: Results for encoder-only models on GLUE dev sets with standard deviations over five runs in parentheses. **Bold** values denote best performing method in each task.

| ATTENTION | BoolQ | CB | RTE | WiC | MultiRC | COPA | WSC | SUPERGLUE Avg. |
|---|---|---|---|---|---|---|---|---|
| SHA | 72.3(0.7) | 88.7(2.5) | 62.5(1.0) | 63.6(0.5) | 59.7(15.7) | 59.2(2.8) | 63.8(1.5) | 67.1(2.4) |
| MHA | 76.6(0.6) | 89.4(1.8) | 67.9(1.3) | 65.4(0.8) | 69.0(1.2) | 64.0(3.1) | 61.5(3.3) | 70.5(0.5) |
| EL-ATT | 73.5(1.0) | 85.7(4.8) | **69.5**(1.4) | 63.8(0.8) | 67.9(0.3) | 62.2(1.8) | 63.8(0.5) | 69.5(1.0) |
| MQA | 74.6(0.6) | 86.7(1.6) | 65.4(0.8) | 64.0(1.2) | **68.8**(0.5) | 62.2(2.0) | 63.3(2.7) | 69.3(0.7) |
| SKV | **75.2**(0.3) | 84.5(3.7) | 67.5(0.9) | **65.2**(1.1) | 68.7(0.2) | **64.0**(1.0) | **64.4**(1.2) | **69.9**(0.4) |
| FNET | 68.4(0.5) | 51.8(4.3) | 60.7(0.9) | 63.8(1.1) | 62.3(0.6) | 58.2(3.7) | 60.0(1.6) | 60.7(0.6) |
| LINEAR | 70.4(0.2) | 50.6(2.1) | 55.2(1.8) | 62.9(0.7) | 57.8(0.5) | 60.0(2.8) | 61.0(1.1) | 59.7(0.9) |
| MHE-ADD | 73.3(0.2) | 88.8(1.7) | 67.5(1.5) | 64.2(0.5) | 67.1(0.2) | 60.2(2.8) | 62.5(1.4) | 69.1(0.5) |
| MHE-MUL | 74.9(0.6) | **89.4**(1.0) | 67.8(1.3) | 64.7(0.6) | 68.0(0.3) | 61.6(1.5) | 61.2(2.9) | 69.6(0.3) |
| MHA(M) | 78.1(0.3) | 88.1(6.8) | 70.3(1.3) | 67.8(0.8) | 72.9(0.6) | 68.2(4.1) | 64.6(2.9) | 72.9(0.6) |
| MHE-MUL (M) | 75.2(0.5) | 84.6(2.4) | 68.6(1.8) | 66.3(0.9) | 69.8(0.4) | 61.6(3.8) | 64.6(0.8) | 70.1(1.1) |

Table 6: Results for encoder-only models on SUPERGLUE dev sets with standard deviations over five runs in parentheses. **Bold** values denote best performing method in each task.

| ATTEN-TION | GLUE | | | | | | | |
|---|---|---|---|---|---|---|---|---|
| | MNLI | QNLI | QQP | RTE | SST | MRPC | CoLA | STS |
| SHA | 80.5 - | 87.5 - | 86.7 - | 63.6 - | 90.7 - | 85.1 - | 53.8 - | 85.8 - |
| MHA | 83.4 (0.02) | 89.8 (0.01) | 87.8 (0.01) | 67.6 (0.03) | 92.0 (0.01) | 86.8 (0.01) | 59.6 (0.05) | 88.5 (0.01) |
| EL-ATT | 81.7 (0.02) | 88.4 (0.02) | 87.3 (0.01) | 67.6 (0.10) | 91.7 (0.02) | 85.9 (0.02) | 52.4 (-0.04) | 87.7 (0.04) |
| MQA | 82.6 (0.04) | 88.8 (0.02) | 87.3 (0.01) | 66.5 (0.06) | 91.4 (0.01) | 87.3 (0.04) | 58.4 (0.12) | 87.9 (0.03) |
| SKV | 82.6 (0.02) | 89.4 (0.02) | 87.7 (0.01) | 68.2 (0.05) | 91.6 (0.01) | 87.4 (0.02) | 56.2 (0.03) | 88.6 (0.02) |
| FNET | 76.3 (-) | 83.8 (-) | 84.8 (-) | 63.2 (-) | 88.4 (-) | 78.0 (-) | 43.2 (-) | 83.7 (-) |
| LINEAR | 75.4 (-) | 81.4 (-) | 85.5 (-) | 54.7 (-) | 90.4 (-) | 72.2 (-) | 50.3 (-) | 70.9 (-) |
| MHE-ADD | 81.5 (3.88) | 87.8 (1.34) | 87.2 (1.86) | 66.9 (16.35) | 90.5 (-0.81) | 87.2 (**7.93**) | 54.7 (5.22) | 87.7 (**7.05**) |
| MHE-MUL | 81.9 (**5.41**) | 87.9 (**1.51**) | 87.4 (**2.54**) | 67.1 (**17.80**) | 91.1 (**1.29**) | 85.4 (1.29) | 56.6 (**16.29**) | 87.3 (5.60) |

Table 7: Detailed average scores and performance elasticity of parameters (in parentheses) on GLUE for MHE models and the baselines with encoder-only architecture using MLM as pre-training objectives. **Underlined** values denote the best performing method and **bold** values denote the method with best PEoP in each task.

| ATTEN-TION | BoolQ | CB | RTE | WIC | SuperGlue
MultiRC | COPA | WSC |
|---|---|---|---|---|---|---|---|
| SHA | 72.3 - | 88.7 - | 62.5 - | 63.6 - | 59.7 - | 59.2 - | 63.8 - |
| MHA | 76.6 (0.03) | 89.4 (0.00) | 67.9 (0.04) | 65.4 (0.01) | 69.0 (0.07) | 64.0 (0.04) | 61.5 (-0.02) |
| EL-ATT | 73.5 (0.03) | 85.7 (-0.06) | 69.5 (0.19) | 63.8 (0.00) | 67.9 (0.23) | 62.2 (0.08) | 63.8 (0.00) |
| MQA | 74.6 (0.04) | 86.7 (-0.03) | 65.4 (0.06) | 64.0 (0.01) | 68.8 (0.21) | 62.2 (0.07) | 63.3 (-0.01) |
| SKV | 75.2 (0.03) | 84.5 (-0.03) | 67.5 (0.06) | 65.2 (0.02) | 68.7 (0.11) | 64.0 (0.06) | 64.4 **(0.01)** |
| FNET | 68.4 (-) | 51.8 (-) | 60.7 (-) | 63.8 (-) | 62.3 (-) | 58.2 (-) | 60.0 (-) |
| LINEAR | 70.4 (-) | 50.6 (-) | 55.2 (-) | 62.9 (-) | 57.8 (-) | 60.0 (-) | 61.0 (-) |
| MHE-ADD | 73.3 (4.58) | 88.8 (0.54) | 67.5 (25.50) | 64.2 (3.00) | 67.1 (39.90) | 60.2 (5.41) | 62.5 (-6.75) |
| MHE-MUL | 74.9 **(11.78)** | 89.4 **(2.52)** | 67.8 **(26.97)** | 64.7 **(5.36)** | 68.0 **(44.63)** | 61.6 **(12.97)** | 61.2 (-13.49) |

Table 8: Detailed average scores and performance elasticity of parameters (in parentheses) on SUPERGLUE for MHE models and the baselines with encoder-only architecture using MLM as pre-training objectives. **Underlined** values denote the best performing method and **bold** values denote the method with best PEoP in each task.

| ATTENTION | MNLI | QNLI | QQP | RTE | SST | MRPC | CoLA | STS | GLUE Avg. |
|---|---|---|---|---|---|---|---|---|---|
| SHA | 78.7(0.1) | 86.0(0.2) | 85.0(0.1) | 66.5(0.9) | 89.8(0.2) | 76.8(0.4) | 38.0(1.3) | 81.5(0.4) | 75.3(0.3) |
| MHA | 80.6(0.1) | 87.9(0.2) | 86.3(0.1) | 66.9(1.1) | 90.2(0.3) | 79.0(0.7) | 42.9(1.3) | 86.0(0.2) | 77.5(0.2) |
| EL-ATT | 79.5(0.2) | 86.8(0.3) | 85.7(0.1) | 65.7(1.4) | 90.0(0.4) | 79.2(1.4) | 41.5(2.2) | 84.3(0.2) | 76.6(0.4) |
| MQA | 80.0(0.1) | 86.3(0.1) | **85.9**(0.1) | 66.2(0.7) | 90.3(0.3) | **80.7**(0.6) | 41.3(0.8) | 84.3(0.4) | 76.9(0.2) |
| SKV | **80.3**(0.1) | **87.5**(0.3) | **85.9**(0.1) | 66.1(1.1) | 90.6(0.5) | 79.6(0.5) | **41.9**(1.8) | **84.9**(0.2) | **77.1**(0.4) |
| MHE-ADD | 78.7(0.1) | 85.6(0.2) | 85.4(0.1) | 66.4(2.5) | 89.6(0.4) | 78.7(0.6) | 38.4(1.2) | 83.5(0.3) | 75.8(0.3) |
| MHE-MUL | 79.0(0.2) | 85.5(0.1) | 85.6(0.1) | **70.2**(2.5) | **90.9**(0.2) | 78.9(0.8) | 39.4(1.3) | 84.0(0.3) | 76.7(0.2) |

Table 9: Results for decoder-only models on GLUE dev sets with standard deviations over five runs in parentheses. **Bold** values denote best performing method in each task.

| ATTEN-TION | MNLI | QNLI | QQP | RTE | GLUE
SST | MRPC | CoLA | STS |
|---|---|---|---|---|---|---|---|---|
| SHA | 78.7 - | 86.0 - | 85.0 - | 66.5 - | 89.8 - | 76.8 - | 38.0 - | 81.5 - |
| MHA | 80.6 (0.01) | 87.9 (0.01) | 86.3 (0.01) | 66.9 (0.00) | 90.2 (0.00) | 79.0 (0.01) | 42.9 (0.06) | 86.0 (0.03) |
| EL-ATT | 79.5 (0.02) | 86.8 **(0.02)** | 85.7 (0.01) | 65.7 (-0.02) | 90.0 (0.01) | 79.2 (0.05) | 41.5 (0.16) | 84.3 (0.06) |
| MQA | 80.0 (0.02) | 86.3 (0.01) | 85.9 (0.01) | 66.2 (-0.01) | 90.3 (0.01) | 80.7 (0.07) | 41.3 (0.12) | 84.3 (0.05) |
| SKV | 80.3 (0.01) | 87.5 (0.01) | 85.9 (0.01) | 66.1 (-0.00) | 90.6 (0.01) | 79.6 (0.03) | 41.9 (0.07) | 84.9 (0.03) |
| MHE-ADD | 78.7 (-0.02) | 85.6 (-1.50) | 85.4 (1.60) | 66.4 (-0.69) | 89.6 (-0.57) | 78.7 (7.96) | 38.4 (3.71) | 83.5 (7.98) |
| MHE-MUL | 79.0 **(0.98)** | 85.5 (-1.88) | 85.6 **(2.05)** | 70.2 **(17.72)** | 90.9 **(4.01)** | 78.9 **(8.58)** | 39.4 **(12.32)** | 84.0 **(9.97)** |

Table 10: Detailed average scores and performance elasticity of parameters (in parentheses) on GLUE for MHE models and the baselines with decoder-only architecture using MLM as pre-training objectives. **Underlined** values denote the best performing method and **bold** values denote the method with best PEoP in each task.

| Hyperparameter | Pretraining |
|---|---|
| Maximum train steps | 1000000 steps |
| Batch size (per GPU) | 32 instances |
| Adam $\epsilon$ | 1e-8 |
| Adam $\beta_1$ | 0.9 |
| Adam $\beta_2$ | 0.9999 |
| Sequence length | 512 |
| Peak learning rate | 1e-4 for MLM |
| Learning rate schedule | linear |
| Warmup steps | 10000 |
| Weight decay | 0.01 |
| Attention Dropout | 0.1 |
| Dropout | 0.1 |

Table 11: Details of hyperparameters used in pre-training.

| Hyperparameter | Fine-tuning |
|---|---|
| Maximum train epochs | 20 epochs for GLUE, SUPERGLUE and SQUAD |
| Batch size (per GPU) | 32 instances |
| Adam $\epsilon$ | 1e-6 |
| Adam $\beta_1$ | 0.9 |
| Adam $\beta_2$ | 0.999 |
| Peak learning rate | 3e-5 for GLUE and SQUAD; 5e-5 for SUPERGLUE |
| Learning rate schedule | cosine with hard restarts |
| Warmup steps | first 6% steps for GLUE and SUPERGLUE; 3327 for SQUAD V1.1; 4950 for SQUAD V2.0 |
| Weight decay | 0 |
| Attention Dropout | 0.1 |
| Dropout | 0.1 |
| Evaluation steps | 2455 for MNLI, 655 for QNLI, 2275 for QQP, 48 for RTE, 421 for SST, 69 for MRPC, 162 for CoLA and 108 for STS, 177 for BoolQ, 5 for CB, 47 for RTE, 102 for WiC, 512 for MultiRC, 8 for COPA, 11 for WSC, 548 for SQUAD V1.1, 815 for SQUAD V2.0 |

Table 12: Details of hyperparameters used in fine-tuning.

| ATTENTION | weights | gradients | Adam states | activations | Total | Memory Saving Ratios (%) |
|---|---|---|---|---|---|---|
| SHA | 4423680 | 4423680 | 5898240 | 25165824 | 39911424 | 44.84 |
| MHA | 14155776 | 14155776 | 18874368 | 25165824 | 72351744 | 0.00 |
| EL-ATT | 7077888 | 7077888 | 9437184 | 25165824 | 48758784 | 32.61 |
| MQA | 7667712 | 7667712 | 10223616 | 25165824 | 50724864 | 29.89 |
| SKV | 10616832 | 10616832 | 14155776 | 25165824 | 60555264 | 16.30 |
| MHE-ADD | 4437504 | 4437504 | 5916672 | 25165824 | 39957504 | 44.77 |
| MHE-MUL | 4437504 | 4437504 | 5916672 | 25165824 | 39957504 | 44.77 |

Table 13: Memory usage (in bytes) and memory saving ratios (compared to MHA) per attention block for our MHE and other baselines. MHA denotes BERT-base here.

| | | #Params(M) | | GLUE | | | SUPERGLUE | | | WIKITEXT-103 | | | PENN TREEBANK | | |
|---|---|---|---|---|---|---|---|---|---|---|---|---|---|---|---|
| | | MHA | MHE-MUL | MHA | MHE-MUL | PRR | MHA | MHE-MUL | PRR | MHA | MHE-MUL | PRR | MHA | MHE-MUL | PRR |
| Encoder | BASE | 28.32 | 8.88 | 81.9 | 80.6 | 98.4 | 70.5 | 69.6 | 98.7 | - | - | - | - | - | - |
| -only | LARGE | 100.66 | 29.96 | 81.5 | 82.6 | 98.7 | 72.9 | 70.1 | 96.2 | - | - | - | - | - | - |
| Decoder | BASE | 28.32 | 8.88 | 77.5 | 76.7 | 99.0 | - | - | - | 43.0 | 53.8 | 74.9 | 44.3 | 50.7 | 85.6 |
| -only | MEDIUM | 100.66 | 29.96 | 79.4 | 77.7 | 97.9 | - | - | - | 35.5 | 37.2 | 95.2 | 37.5 | 41.6 | 88.5 |

Table 14: Results of evaluation metrics on two text classification benchmarks (GLUE, SUPERGLUE) and two language modelling benchmarks (WIKITEXT-103 and PENN TREEBANK) with performance retention ratio (PRR) for MHA and MHE-MUL across different model sizes.

| | N | $d_m$ | h | $d_h$ | $p_{drop}$ | #Steps | #Params(M) | | BLEU | | PRR |
|---|---|---|---|---|---|---|---|---|---|---|---|
| | | | | | | | MHA | MHE-MUL | MHA | MHE-MUL | |
| BASE | 12 | 512 | 8 | 64 | 0.1 | 100K | 18.87 | 6.52 | 24.8 | 23.6 | 95.0 |
| | 12 | 512 | 16 | 32 | 0.1 | 100K | 18.87 | 5.63 | 25.1 | 22.9 | 91.5 |
| | 12 | 512 | 4 | 128 | 0.1 | 100K | 18.87 | 8.29 | 24.7 | 23.6 | 95.3 |
| 4L | 8 | 512 | 8 | 64 | 0.1 | 100K | 12.58 | 4.34 | 23.9 | 22.4 | 93.6 |
| 8L | 16 | 512 | 8 | 64 | 0.1 | 100K | 25.17 | 8.69 | 25.3 | 24.3 | 96.0 |
| 12H | 12 | 768 | 12 | 64 | 0.15 | 100K | 42.47 | 13.31 | 25.7 | 24.2 | 94.2 |
| BIG | 12 | 1024 | 16 | 64 | 0.3 | 300K | 75.50 | 22.47 | 26.5 | 24.8 | 93.6 |

Table 15: Results of BLEU scores on WMT-14 English to German machine translation task with performance retention ratio (PRR) for MHA and MHE-MUL across different model sizes.