# OpenReview forum: "Pit One Against Many: Leveraging Attention-head Embeddings for Parameter-efficient Multi-head Attention"
_EMNLP/2023/Conference — EMNLP 2023 Findings_

### Official Review · Reviewer_M1Z9 · 2023-07-22

**Soundness:** 3

**Excitement:**

4: Strong: This paper deepens the understanding of some phenomenon or lowers the barriers to an existing research direction.

**Paper Topic And Main Contributions:**

This paper proposes a memory efficient method to build the attention module in Transformers, which shows a high performance retention in many natural language understanding tasks and one translation task compared to the vanilla multi-head attention module of encoder only and encoder-decoder models. The contributions of this paper lie in the approaches for compute efficiency and NLP engineering experiments.

**Questions For The Authors:**

1. It seems that MHE performs worse in translation task (92%~95% in PRR) than the other natural language understanding tasks tested (most of them are greater than 95%). Are there any more results in other natural language generation tasks, e.g., Text Summarization and Question-Answering tasks, to support the effectiveness of your method?

2. Why not conduct additional experiments to further verify the effectiveness of MHE in parameter utilization? For example, under the same parameters used settings, the parameters saved in MHA sub-layer are re-invested to the other sub-layers, e.g., the following feed-forward network, to compare with the MHA baseline.

3. What are the results of models in Table 1 on the corresponding test set?

**Reasons To Accept:**

- The proposed method, Multiple Head Embedding module (MHE), reduces one order of memory needed in the attention sub-layer of Transformers, while preserving more than 90% performance of vanilla multi-head attention module in the downstream tasks tested.
- According to the results in this paper, it is interesting to be investigated further under the large language model settings.

**Reasons To Reject:**

- Experiments of decoder-only models and many natural language generation tasks are missing while this paper aims to mitigate the computational problems in the scaling language models, which are mainly decoder only models (e.g., GPT-3, LLaMA, and GLM).

Minor:
- Test set results are more compelling and able to show the actual performance of models than the dev set results in Table 1.

- In transformer layer, most of parameters are contained in the feed-forward networks (≈2/3) rather than multi-head attention sub-layers (≈1/3). Thus the method in this paper, focusing the memory-efficient attention sub-layer, has less significance compared to the one in feed-forward network.

**Reproducibility:**

4: Could mostly reproduce the results, but there may be some variation because of sample variance or minor variations in their interpretation of the protocol or method.

**Reviewer Confidence:**

4: Quite sure. I tried to check the important points carefully. It's unlikely, though conceivable, that I missed something that should affect my ratings.

**Typos Grammar Style And Presentation Improvements:**

- Line 21 in the abstract: (3n^2-3n)d^2 --> (3n^2-3n)d^2-3nd (-3nd is missing)

---

> ### Author Rebuttal · Authors · 2023-08-29
>
> **Response to “Focusing the memory-efficient attention sub-layer, has less significance compared to the one in feed-forward network”**: Same as the response to the comment by Reviewer ymVs:  The parameters of MHA occupy 1/3 of the whole parameters in the encoder and 1/2 in the decoder Transformer layers respectively. On the other hand, the Feed-forward sublayers (FFNs) occupy 2/3 in the encoder and 1/2 in the decoder layers respectively. For example, in a  BERT-base model, our approach could substantially save approximately 18% of the whole model parameters (given that the parameters in the embedding matrix account for the 20%). We admit that the parameter size of the FFN in the Transformer is also significant. Reducing the FFN size does not conflict with our parameter-efficient MHE attention. We opted to use the conventional default settings (intermediate_size = 4 * hidden_size) for all models tested for simplicity and also keeping the number of experiments manageable given our limited compute. Our intuition is that reducing the size of the FFN would reduce model performance but it is something that could be tested in future work.
>
> We also provide responses to each of your questions as follows:
>
> 1. **Experiments on Decoder-only Architecture**: We extended experiments by pretraining GPT2 models, both in base and large sizes, using a 512-token context. The results of our evaluation on general text classification tasks are as follows:
> - Base-sized MHE-Mul (12 heads, 12 decoder layers) achieved a Performance Retention Ratio (PRR) of 99.0% on the GLUE Avg. score.
>
> - Medium-sized MHE-Mul (16 heads, 24 decoder layers) attained a PRR of 97.9%.
>
> We will add this in the camera-ready upon acceptance given the extra space.
>
> We also report model perplexities:
>
> |            |         | perplexity   |               |
> | ---------- | -------:| ------------:| -------------:|
> | Model      | #params | WikiText-103 | Penn Treebank |
> | SHA        | 8.85M   | 62.0         | 68.1          |
> | MHA        | 28.32M  | 43.0         | 44.3          |
> | EL-Att     | 14.16M  | 57.1         | 56.1          |
> | MQA        | 15.34M  | 49.7         | 49.3          |
> | SKV        | 21.23M  | 46.2         | 45.5          |
> | MHE-Add    | 8.88M   | 54.0         | 55.3          |
> | MHE-Mul    | 8.88M   | 53.8         | 50.7          |
> | MHA(M)     | 100.66M | 35.5         | 37.5          |
> | MHE-Mul(M) | 29.96M  | 37.2         | 41.6          |
>
> Although we have not conducted experiments on text summarization tasks due to the resource-intensive nature of pre-training with longer contexts, we think that this can be addressed in future work. We believe that our new experiments together with those included in the original submission demonstrate that our proposed MHE exhibits great parameter efficiency (PEoP scores)  and consistently achieves good performance retention ratios.
>
> 2. **Scope of the Study and Feed-forward Layers**: The scope of our present study centres on enhancing the parameter-efficiency of the multi-head attention mechanism by introducing attention-head embeddings. While we acknowledge the potential for parameter savings within feed-forward layers, our focus remained on the design and efficacy of the attention mechanism. The interplay between our approach and parameter-efficient feed-forward sublayers falls within the scope of future research, as we anticipate that these strategies can seamlessly complement each other. See our response above.
>
> 3. **Ground Truth Labels and Benchmark Protocols**: We operated under the same protocol as established by previous work such as the experimental design of RoBERTa and DistilBERT papers. Given that the ground truth labels in the test datasets are inaccessible, we adhered to these established protocols to ensure consistency and comparability with prior research.
>
> We trust that our responses provide comprehensive insights into our methodology, scope, and results in the context of the questions raised.

---

### Official Review · Reviewer_ymVs · 2023-07-30

**Soundness:** 3

**Excitement:**

2: Mediocre: This paper makes marginal contributions (vs non-contemporaneous work), so I would rather not see it in the conference.

**Paper Topic And Main Contributions:**

This paper proposes a parameter-efficient method for multi-head attention. The standard multi-head attention provides weight matrices corresponding to the number of attention heads. In contrast, the proposed method provides only one weight matrix for key, value, and query in the attention. To modify each key, value, and query, the proposed method uses weight vectors corresponding to the number of heads. In the modification, the authors provides two approaches, addition and multiplication. The proposed method outperformed the single-head attention method though the number of parameters is almost the same.

**Questions For The Authors:**

In my understanding, the proposed method can be combined with the previous approaches such as MQA and SKV to increase the expressiveness of these methods. Did the authors conduct such experiments?

**Reasons To Accept:**

The proposed method is simple but effective. It consistently achieved better performance than the single-head attention.

In my understanding, the proposed method can be combined with previous parameter-efficient method. As in the question, the authors did not investigate the effectiveness but the proposed method might improve the performance.

**Reasons To Reject:**

The proposed method achieved better performance the single-head attention, but I cannot agree that the proposed method is more useful than the existing parameter-efficient attention methods such as multi-query attention. In fact, MQA and SKV consistently achieved better performance than the proposed method.
In addition, I wonder about the baseline performance in the machine translation experiments. In WMT English-to-German 2014 dataset, we can achieve 27 BLEU score with the standard (i.e., MHA) Transformer. If the authors did not use SacreBLEU to compute the BLEU score, the standard Transformer achieve around 28 BLEU score as in [Takase and Kiyono, 2023]. In short, I suspect that the baseline is weak in comparison to the widely used configuration.

In my understanding, the proposed method is parameter-efficient but not memory-efficient because the space complexity of the proposed method is almost the same as the one of the multi-head attention during the attention calculation. I recommend the authors indicate the saving rate of the memory. In addition, I wonder if the parameters of multi-head attention are dominant in parameters of the Transformer. The authors show the parameter size on attention mechanisms in the experiments but I'm not sure the proposed method has a significant effect on the whole parameter size.


Takase and Kiyono, 2023: Lessons on Parameter Sharing across Layers in Transformers.

**Reproducibility:**

4: Could mostly reproduce the results, but there may be some variation because of sample variance or minor variations in their interpretation of the protocol or method.

**Reviewer Confidence:**

4: Quite sure. I tried to check the important points carefully. It's unlikely, though conceivable, that I missed something that should affect my ratings.

---

> ### Author Rebuttal · Authors · 2023-08-29
>
> We thank you for your constructive feedback.
>
> **Balance between Predictive Performance and Memory**: We greatly appreciate the reviewer ymVs's attention to the pursuit of high predictive performance. It's our paramount goal to balance the model parameter efficiency with its predictive capabilities. While it's true that preceding parameter-sharing techniques such as SKV and MQA displayed marginally better outcomes, it's noteworthy that these methods demand an exponential increase in parameters relative to the number of heads employed. This equilibrium between parameter efficiency and performance is encapsulated by what we term "performance elasticity of parameters" (PEoP) which indicates  how effectively the model parameters contribute to predictive performance. By measuring PEoP, our proposed method achieves state-of-the-art performance, indicating that it obtains substantially higher predictive performance per parameter (i.e. more parameter efficient). It's worth highlighting that our attention models can be further integrated with SKV and MQA, and we plan to explore this in future work. Our methods can be used under limited memory budgets, achieving high performance retention ratios across a diverse set of tasks.
>
> **Main Contributions**: We would like to emphasise that the novelty of our work lies in leveraging multiple head-embeddings with a single shared projection matrix to support multiple attention heads. Our method (MHE) only requires a very small fraction of additional parameters compared to a single-head attention mechanism but achieves comparable predictive performance on a variety of downstream tasks to full multi-head attention (MHA).
>
> **Why not combine MHE with other efficient attention methods?**: Applying MHE with other parameter-efficient attention methods (e.g. MHQ/SKV, parameter-efficient embeddings etc.) is left for future work. The main focus of our paper is to demonstrate the parameter efficiency and robustness in performance retention ratio of vanilla MHE compared to vanilla MHA.
>
> **Are the parameters of multi-head attention dominant in the parameters of the Transformer?**: The parameters of MHA occupy 1/3 of the whole parameters in the encoder and 1/2 in the decoder Transformer layers respectively. On the other hand, the Feed-forward sublayers (FFNs) occupy 2/3 in the encoder and 1/2 in the decoder layers respectively. For example, in a  BERT-base model, our approach could substantially save approximately 18% of the whole model parameters (given that the parameters in the embedding matrix account for the 20%). We admit that the parameter size of the FFN in the Transformer is also significant. Reducing the FFN size does not conflict with our parameter-efficient MHE attention. We opted to use the conventional default settings (intermediate_size = 4 * hidden_size) for all models tested for simplicity and also keeping the number of experiments manageable given our limited compute. Our intuition is that reducing the size of the FFN would reduce model performance but it is something that could be tested in future work.
>
> **Parameter-efficiency ~= Memory-efficiency**: Our understanding is that parameter-efficiency mirrors memory efficiency. For example, when leveraging mixed-precision training with float16 and the Adam optimizer, each parameter needs 20 bytes of memory: 2 bytes for the weights, 4 bytes for the gradients, and an additional 4 bytes for Adam states, summing up to 20 bytes per parameter. Notably, this alignment between parameter-efficiency and memory consumption has been highlighted in Section 2.1.1 of the work titled "Using DeepSpeed and Megatron to Train Megatron-Turing NLG 530B, A Large-Scale Generative Language Model" (Smith et al., 2022).
>
> **BLEU Calculation Details**: We deeply appreciate the reviewer's valid concerns regarding the computation of BLEU scores which we have updated below. We noticed an issue was present within the GitHub repository we initially referenced, specifically in the file /microsoft/DirectML/blob/master/PyTorch/1.13/attention_is_all_you_need/transformer/Models.py#173/blob/master/PyTorch/1.13/attention_is_all_you_need/transformer/Models.py#173. Subsequently, we conducted a comprehensive reevaluation of our experiments using an alternative repository, namely /sgrvinod/a-PyTorch-Tutorial-to-Machine-Translation, employing the default hyperparameters settings. We employ the official WMT-calculated BLEU score (sacreBLEU signature: BLEU+case.mixed+lang.en-de+numrefs.1+smooth.exp+test.wmt14/full+tok.13a+version.1.4.3) as our benchmark for evaluation. Our findings were subsequently updated and are presented in the revised table. The BLEU score reported in the original paper stands at 27. We acknowledge that this figure might not align with calculations performed using different methods, as evidenced by discussions in Issues comments 317#issuecomment-377580270 and 317#issuecomment-380970191 within the official tensor2tensor repository by TensorFlow. Throughout our experimentation, we have maintained uniformity in hyperparameter settings across all models, thereby illustrating their relative BLEU score differences. Please note that the different calculation of BLEU did not change the relative differences in performance across methods.
>
> |         | #params(att) | Avg. | PRR   |
> | ------- | ------------:| ----:| -----:|
> | SHA     | 6.49M        | 22.5 | 90.8  |
> | MHA     | 18.87M       | 24.8 | 100.0 |
> | EL-Att  | 9.44M        | 23.9 | 96.6  |
> | MQA     | 10.62M       | 24.2 | 97.6  |
> | SKV     | 14.16M       | 24.7 | 99.5  |
> | MHE-Add | 6.52M        | 23.0 | 92.9  |
> | MHE-Mul | 6.52M        | 23.6 | 95.0  |
>
> We will update Table 2 with the new results in the camera ready version upon acceptance.

---

### Official Review · Reviewer_oMfC · 2023-07-30

**Soundness:** 3

**Excitement:**

3: Ambivalent: It has merits (e.g., it reports state-of-the-art results, the idea is nice), but there are key weaknesses (e.g., it describes incremental work), and it can significantly benefit from another round of revision. However, I won't object to accepting it if my co-reviewers champion it.

**Paper Topic And Main Contributions:**

This paper proposes to use one matrix for all query, key, value projections for all layers in Transformer and add/multiply with head embeddings. They show that their model achieves better performance than single headed attention (which has similar number of parameters) and is comparable to other memory saving approaches while saving more memory. They show that their approach does not increase a lot of memory when scaling up the number of attention parameters. They also show that their model has better parameter utilization than previous works.

**Questions For The Authors:**

A. How is the performance changing when scaling up the number of parameters?

**Reasons To Accept:**

1. The approach is simple and intuitive, and improves memory efficiency in Transformer attention.
2. Comparing to models with similar number of params, their approach gives better performance.

**Reasons To Reject:**

1. The paper claims good memory efficiency when scaling up, but does not show how performance is affected when scaling up. So it is unclear that its performance will not degrade too much when scaling up.

**Reproducibility:**

4: Could mostly reproduce the results, but there may be some variation because of sample variance or minor variations in their interpretation of the protocol or method.

**Reviewer Confidence:**

4: Quite sure. I tried to check the important points carefully. It's unlikely, though conceivable, that I missed something that should affect my ratings.

---

> ### Author Rebuttal · Authors · 2023-08-29
>
> We deeply appreciate your insightful feedback.
>
> **Additional Results towards Scaling**: It's important to mention that our experimentation was conducted under the constraints of limited computational resources, which naturally hindered our ability to pre-train models on a very large scale. Despite these inherent limitations, we have performed supplementary experiments that involve comparing the vanilla Multi-Head Attention and our best performing MHE-Mul approach on different scales by adjusting the number of attention heads.
>
> Specifically, we experimented with 12 and 16 heads per encoder and decoder layer (in the paper we only report results using 8 heads) on the WMT14 machine translation task. We observed comparable performance retention ratios for MHE-Mul when scaling to 12 and 16 heads of 94.2% and 93.6% respectively (similar to the results obtained with 8 heads per layer reported in the paper). Moreover, we extended our experiments by varying the numbers of layers in the encoder-decoder model, i.e. 8 and 16 layers in total (4 and 8 layers for both the encoder and the decoder respectively) from 12 that we have reported in the paper. Similarly, the performance retention ratios are 93.6% for 8 layers and 96.0% for 16 layers with MHE-Mul.
>
> We have also pre-trained a BERT-large model using MHE-Mul (16 heads and 24 layers). MHE-Mul yielded a SuperGlue Avg. score of 70.1, in comparison to the 72.9 achieved by using MHA in BERT-large, resulting in a performance retention ratio of 96.2%.
>
> Collectively, these additional results demonstrate the high and robust performance retention ratio of MHE-Mul across different model sizes by varying the number of attention heads and layers. We hope that these additional experiments and results address your concerns regarding the generalizability of our parameter efficient attention given the constraints in computing resources. We will include all these extra results in the camera ready version given the extra space.
>
> |                | #params(att) | SuperGlue Avg. | PRR   |
> | -------------- | ------------:| --------------:| -----:|
> | MHA(base)      | 28.32M       | 70.5           | 100.0 |
> | MHE-Mul(base)  | 8.88M        | 69.6           | 98.7  |
> | MHA(large)     | 100.66M      | 72.9           | 100.0 |
> | MHE-Mul(large) | 29.96M       | 70.1           | 96.2  |

---

### Meta-Review · Area_Chair_KQtp · 2023-09-11

**Recommendation:** 3

**Metareview:**

This paper proposes to save space in transformers by using a single matrix for all query, key, value projections for all layers. They compare their approach to various baselines, showing memory savings. Reviewers found the approach simple and intuitive, and liked that it can be combined with other methods. They also generally found the results strong, but  complained about missing details in the reporting, and about the quality of the baselines. The rebuttal helped alleviate some of these concerns, though the baseline concern still remains.

---

### Decision · Program_Chairs · 2023-10-07

**Decision:**

Accept-Findings

**Comment:**

This paper proposes to save space in transformers by using a single matrix for all query, key, value projections for all layers. They compare their approach to various baselines, showing memory savings. Reviewers found the approach simple and intuitive, and liked that it can be combined with other methods. They also generally found the results strong, but  complained about missing details in the reporting, and about the quality of the baselines. The rebuttal helped alleviate some of these concerns, though the baseline concern still remains.